# Real-time monitoring of replication errors' fate reveals the origin and dynamics of spontaneous mutations

Chiara Enrico Bena[1], Jean Ollion [2,3], Marianne De Paepe[1], Magali Ventroux [1], Lydia Robert [1,4] ✉ & Marina Elez [1,4] ✉

The efficiency of replication error repair is a critical factor governing the emergence of mutations. However, it has so far been impossible to study this efficiency at the level of individual cells and to investigate if it varies within isogenic cell populations. In addition, why some errors escape repair remains unknown. Here we apply a combination of fluorescent labelling of the *Escherichia coli* Mismatch Repair (MMR) complex, microfluidics, and time-lapse microscopy, to monitor in real-time the fate of >20000 replication errors. We show that i) many mutations result from errors that are detected by MMR but inefficiently repaired ii) this limited repair efficiency is due to a temporal constraint imposed by the transient nature of the DNA strand discrimination signal, a constraint that is likely conserved across organisms, and iii) repair capacity varies from cell to cell, resulting in a subpopulation of cells with higher mutation rate. Such variations could influence the fitness and adaptability of populations, accelerating for instance the emergence of antibiotic resistance.

Many spontaneous mutations result from replication errors[1,2]. Errors of the DNA polymerase during replication lead to mismatched bases or small insertion or deletion loops, that can be repaired by a dedicated system called Mismatch Repair (MMR)[3]. Therefore, mutations occur in two steps: the production of an erroneous DNA sequence by the polymerase, and the failure of its repair by MMR.

MMR is widely conserved in the whole tree of life[4] and has been deciphered in detail in *Escherichia coli*, where it involves a dozen of proteins, most importantly MutS, MutL, and MutH[5,6]. The detection of errors is performed by MutS, and the strand discrimination task is performed by the MutH endonuclease, which cleaves the newly synthesized strand at hemi-methylated GATC sites[5,6]. In *E. coli*, adenine in GATC sequences is methylated by the Dam methyltransferase[7]. Methylation by Dam typically occurs a few minutes after replication and therefore the newly synthesized strand is transiently unmethylated after replication[8,9]. MutH cleaves the unmethylated strand, thus directing repair to the newly synthesized strand[5,6]. The third major player of MMR, the MutL protein, is a matchmaker that coordinates the different steps of MMR[10].

MMR is very efficient and ensures high replication fidelity in all organisms studied to date. In *E. coli* it allows repairing >99% of errors, resulting in a remarkable ~150-fold reduction in the rate of spontaneous mutations[2]. Despite extensive research on MMR spanning over four decades, the reasons why 1% of errors are not repaired and therefore lead to mutations remain unknown[11,12]. It is also unknown whether MMR failures occur randomly, with a rate that is constant through time and homogeneous within an isogenic cell population, or if they occur in cells that are in a transient state of impaired repair capacity[13,14]. Cells with transiently impaired repair capacity could lead to bursts of mutations that could have important evolutionary consequences, impacting the overall fitness and adaptation capacity of the population[15].

[1]Université Paris-Saclay, INRAE, AgroParisTech, Micalis Institute, 78350 Jouy-en-Josas, France. [2]Sorbonne Université, CNRS, Institut de Biologie Paris-Seine (IBPS), Laboratoire Jean Perrin (LJP), 75005 Paris, France. [3]SABILab, Die, France. [4]These authors jointly supervised this work: Lydia Robert, Marina Elez. ✉e-mail: Lydia.robert@inrae.fr; Marina.elez@inrae.fr

We previously developed an imaging approach to characterize the first step of spontaneous mutation occurrence, i.e., error production[16–18] (Fig. 1a–c), directly in *E. coli* single cells. It combines microfluidics, widefield fluorescence microscopy, and a fluorescently tagged YFP-MutL. In MMR-deficient *mutH* cells, this protein forms fluorescent foci on replication errors (Fig. 1b, c). Since all errors in *mutH* cells are converted into mutations, the fluorescent foci persist on DNA until a new replication cycle transforms the error into a mutation, i.e., on average a doubling time (Fig. 1b), and the rate of appearance of fluorescent foci is equal to the mutation rate[17]. Using this approach, we showed that in cells growing in non-stressful conditions, errors are produced at a constant rate, i.e., replication fidelity is homogeneous within an isogenic population[17].

Here, we characterize the second and last step controlling the appearance of mutations, i.e., error repair, in *E. coli*. We first develop an experimental approach to study the efficiency of repair at the level of individual cells, based on the visualization of fluorescently tagged YFP-MutL in MMR-proficient cells (Fig. 2). To validate this approach we show that (i) unrepaired and repaired errors can both be visualized, which was unclear from previous studies[12,17,19,20], and (ii) they can be distinguished based on the lifetime of the fluorescent foci. We use this visualization strategy with cells growing in a mother machine micro-fluidic chip, which is composed of thousands of narrow microchannels where cells grow in a single file (Fig. 1a). The microchannels are closed on one side and open on the other to a large channel through which the growth medium flows, ensuring constant renewal of the medium. As bacteria grow and fill the microchannels, those that are pushed out are carried away by the flow of the medium. This setup therefore ensures stable growth conditions on a long timescale. In order to study repair efficiency at the single-cell level, we performed time-lapse experiments where ~400 microchannels were imaged every 2 min for ~30 h (~70 cell divisions), leading to the detection of ~7000 replication errors (Supplementary Table 1). Tracking and quantifying fluorescent foci and analyzing their lifetimes provides insights into single-cell repair capacity, the molecular mechanisms underlying repair failures, and reveals the dynamics of mutations in repair-proficient cells.

## Results

### The rate of YFP-MutL foci matches the rate of replication errors in WT cells

We first tested whether YFP-MutL imaging can be used to detect all errors in repair-proficient cells and distinguish between repaired and unrepaired ones. To do so, we first quantified the rate of YFP-MutL foci occurrence in a WT strain by performing short-term (~4 min) time-

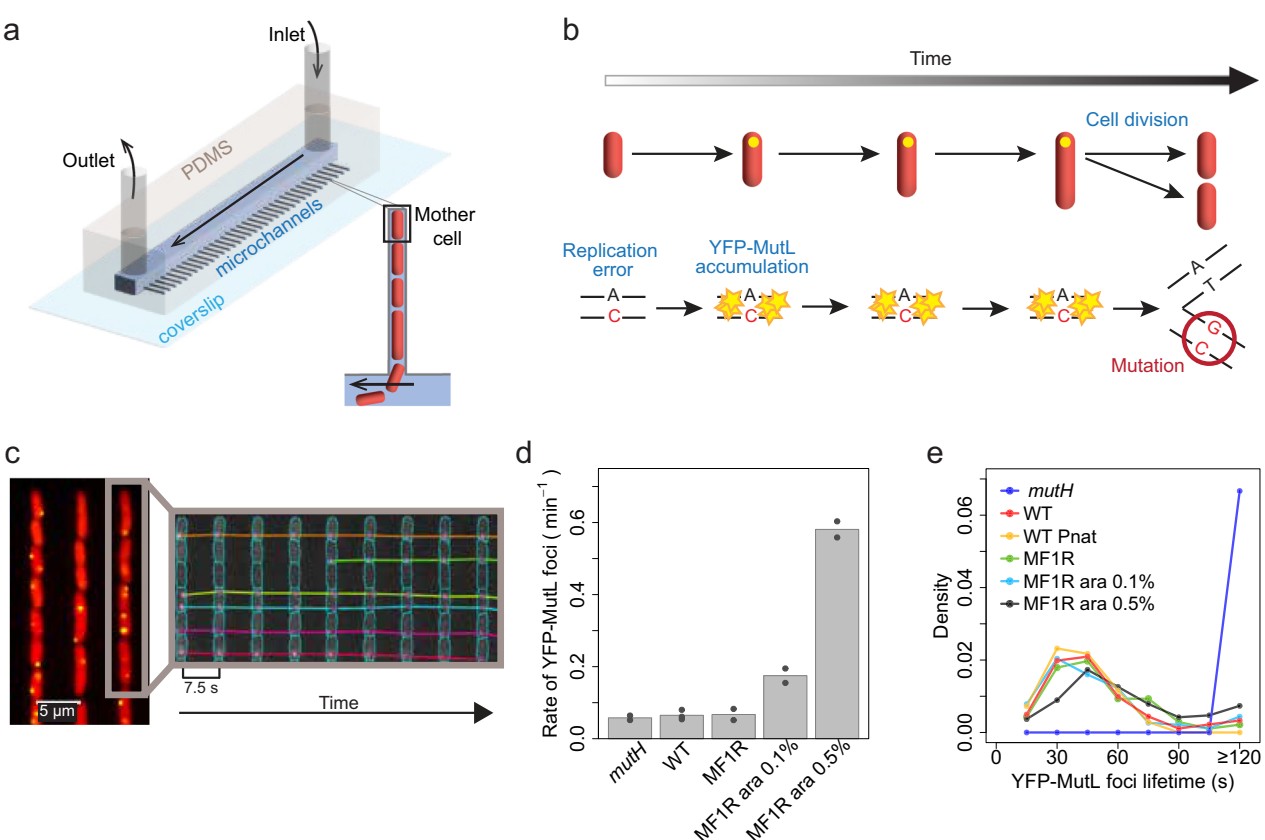

**Fig. 1 | Following YFP-MutL foci in *E. coli*. a** The mother machine microfluidic chip contains ~ a thousand narrow microchannels in which bacteria grow in a single file. Modified from Robert et al.[17] with permission from AAAS and the authors. **b** Tracking replication errors with fluorescent YFP-MutL in *mutH* cells (depicted in red). The fluorescent YFP-MutL focus (yellow dot) remains associated with DNA until a subsequent replication cycle converts the error into a mutation. Modified from Robert et al.[17] with permission from AAAS and the authors. **c** Representative fluorescent image (left) of *mutH* cells expressing YFP-MutL and the corresponding kymograph (right). One representative experiment out of 2 performed. The kymograph obtained by BACMMAN software displays the segmentation and tracking over time (time interval of 7.5 s) of cells (cyan contours) and YFP-MutL foci (magenta contours for segmentation and colored lines for tracking) within a single microchannel (gray rectangle). **d** Rates of YFP-MutL foci in *mutH*, wild-type (WT), and MF1R strain under different levels of induction of the $P_{BAD}$ promoter (no induction, arabinose 0.1% and 0.5%). The bars represent the average of individual experiments (dots). See the first section of Supplementary Note 1 for details, Supplementary Table 2 for the values of individual experiments, and Supplementary Table 3 for comparisons of the different conditions. **e** The distribution of YFP-MutL foci lifetimes of one representative experiment per condition (all experiments are shown in Supplementary Fig. 1, see also Supplementary Table 4 for the total number of experiments performed and foci analyzed). Source data are provided as a Source Data file.

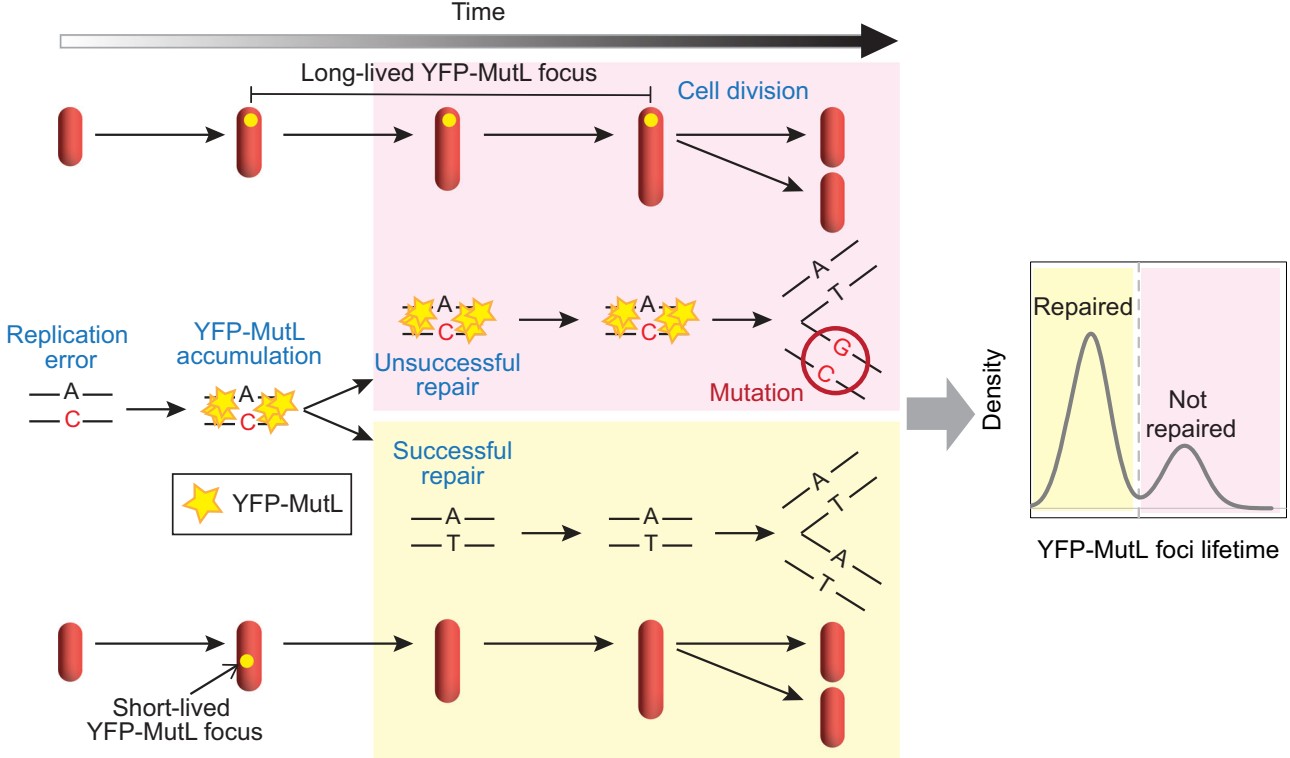

**Fig. 2 | Experimental approach to study the efficiency of error repair at the single-cell level.** A replication error that has been detected by MMR can either be repaired, leading to a short-lived YFP-MutL focus (bottom and yellow-shaded part of the schematic), or converted into a mutation, resulting in a long-lived focus (top and pink-shaded part of the schematic). YFP-MutL foci are represented as yellow dots in red cells. Modified from Robert et al.[17] and Elez[43] with permission from AAAS and the authors.

lapse experiments (see Methods section on Microscopy). Cells grown in LB medium in the mother machine microfluidic chip (Fig. 1a) and expressing YFP-MutL were monitored every 7.5 or 15 seconds using widefield fluorescence microscopy. For comparison, the same experiments were performed with repair-deficient *mutH* cells, where all replication errors are detected[17]. Images were analyzed using BACMMAN software[16,21] (see Methods section on Image analysis) to detect, track, and quantify YFP-MutL foci (Fig. 1c). The rate of fluorescent foci in the WT strain matches that of the *mutH* strain (0.065 min$^{-1}$ vs. 0.058 min$^{-1}$, respectively) (Supplementary Tables 2 and 3 and Fig. 1d). Because in WT cells ~99% of errors are successfully repaired[2], this suggests that foci in the WT strain mostly correspond to replication errors undergoing faithful repair.

## Short-lived YFP-MutL foci represent repaired errors in WT cells

To confirm that most foci in WT cells mark faithful repair of replication errors, we quantified their lifetimes. MutH cannot cleave fully methylated DNA. Therefore, the transient wave of DNA hemimethylation following replication creates a window of opportunity for MMR, which is on the order of minutes[8,9]. Thus, repaired errors should lead to short-lived YFP-MutL foci compared to foci in the MMR-deficient *mutH* strain, which last on average one doubling time (Fig. 1b)[17]. We found that in WT cells, where 99% of errors are repaired, fluorescent foci have an average lifetime of ~ 40 s (Fig. 1e and Supplementary Fig. 1). The frequent illumination of YFP-MutL in these experiments leads to substantial photobleaching (Supplementary Fig. 2a). However, under the same conditions, the foci of the *mutH* strain last much longer than those of the WT strain: 99.6% of foci last at least 2 min in the *mutH* strain, whereas this proportion drops to 1.8% in the WT strain (Fig. 1e and Supplementary Fig. 1). This shows that WT short-lived foci are not the result of photobleaching and confirms that WT foci have a typical

lifetime shorter than 1 min. This short lifetime is in agreement with the hypothesis that they correspond to repaired errors.

To strengthen this conclusion, we modulated the error rate using *dnaQ926*, a deficient allele of the Pol III polymerase proof-reading subunit[17]. This dominant negative allele was introduced under the control of an inducible promoter (strain MF1R). We first estimated the rate of errors in MF1R at different induction levels, using the classical approach of mutation accumulation followed by whole genome sequencing (MA + WGS, see Methods section on MA + WGS and Supplementary Note 2). MA + WGS allows estimating the rate of mutation. Therefore, in order to estimate the replication error rate in the MF1R strain, we inactivated *mutS*, allowing all replication errors to give rise to mutations and thus to be detected by MA + WGS. We show in Supplementary Fig. 3 and Supplementary Table 5 that in the MF1R*mutS* strain, the mutation rate, i.e., the error rate, increases progressively with inducer concentration. Figure 1d and Supplementary Table 2 show that the rate of appearance of YFP-MutL foci also increases in cells when MF1R is induced. In addition, the distribution of foci lifetime in cells with disrupted Pol III proof-reading activity is similar to the distribution of foci lifetime in WT cells (Fig. 1e and Supplementary Fig. 1). Altogether, these results show that short-lived YFP-MutL foci in WT *E. coli* mark errors that are successfully repaired.

## Successful and unsuccessful repair events can be tracked simultaneously and quantified in vivo

Failures of the MMR system could occur either at the detection or at the repair step. If an error escapes detection by MMR, it cannot be visualized with YFP-MutL, in contrast to events of repair failures. If the error is detected but repair is not completed during the window of opportunity created by the transient wave of DNA hemimethylation, MutH will not cleave DNA and repair will fail. Such errors, that are

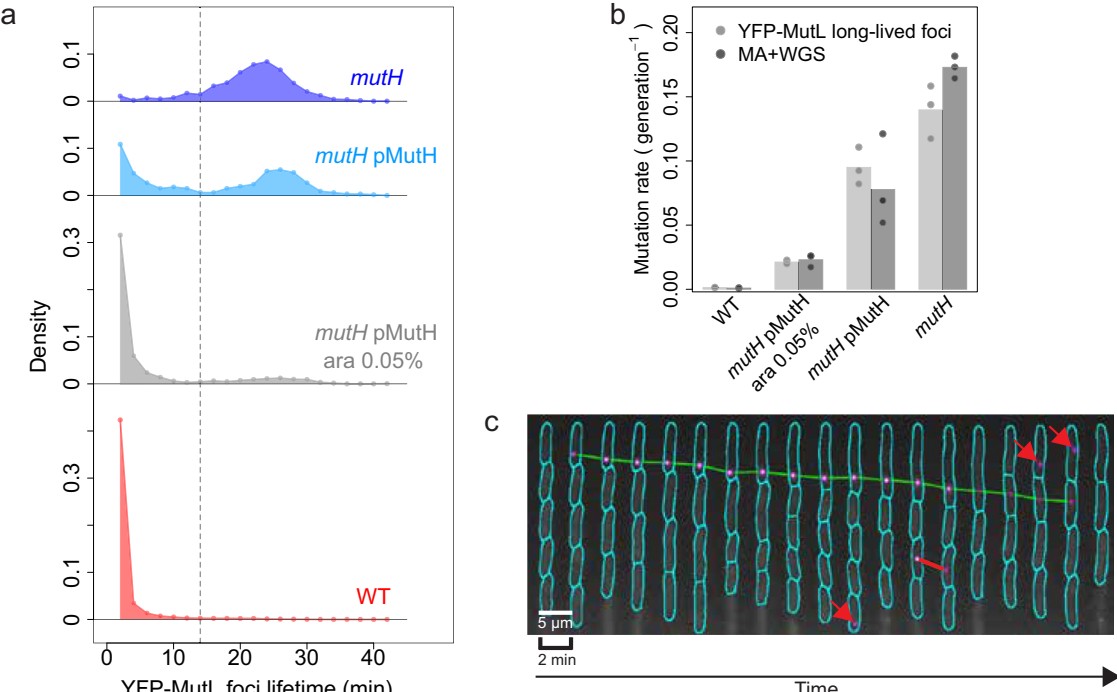

**Fig. 3 | Detection and quantification of successful and unsuccessful repair of replication errors. a** Distributions of YFP-MutL foci lifetimes in strains with varying repair efficiency: *mutH* cells (in blue, 748 foci), *mutH* cells expressing MutH from the P_BAD promoter on a plasmid (*mutH* pMutH), grown either in LB (*mutH pMutH* in cyan, 341 foci) or in LB supplemented with 0.05% of arabinose (*mutH pMutH* ara 0.05% in gray, 1190 foci), and WT cells (in red, 6241 foci). The dashed gray line represents the threshold at 14 min, which separates short-lived foci (on the left) from long-lived foci (on the right). Data presented in the figure are from one representative experiment for each condition. The total number of experiments performed for each condition are: 3 for *mutH*, WT, and *mutH* pMutH; 2 for *mutH* pMutH ara 0.05%. **b** Estimates of the mutation rate obtained using two approaches: long-lived YFP-MutL foci (lifetime >14 min) (light gray, total number of experiments is the same as in panel (**a**), and mutation accumulation + whole genome sequencing experiment (MA + WGS, dark gray) for the same conditions as shown in panel (**a**). The bars represent the average of individual experiments (dots, n = 3 for *mutH* and *mutH* pMutH ara 0.05%, n = 4 for *mutH* pMutH and n = 2 for WT). The values of MA + WGS for the WT strain are from Lee et al.[2]. **c** Example of a kymograph of WT cells growing in one microchannel over a 38-min period, showing foci tracks (green and red lines; foci appearing on a single frame are indicated by red arrows). In total 3 WT experiments have been performed with similar results. Source data are provided as a Source Data file.

detected but unsuccessfully repaired should therefore lead to YFP-MutL foci with a lifetime similar to that observed in the MMR-deficient *mutH* strain. Our visualization method should therefore enable successful and unsuccessful repair events to be detected as YFP-MutL foci, and distinguished according to their lifetime.

To demonstrate that, we modulated repair efficiency by changing the concentration of MutH. We performed long-term (between 17 and 46 h) time-lapse experiments where YFP-MutL foci occurrences are monitored every 2 min using a WT strain or a strain where MutH is expressed on a plasmid under the control of the arabinose-inducible promoter P_BAD while the native *mutH* is deleted (*mutH* pMutH strain). Figure 3a shows the foci lifetime distributions for the WT and *mutH* strains, as well as for the *mutH* pMutH strain growing in LB medium with or without 0.05% arabinose. When grown in the absence of arabinose, the leakiness of the P_BAD promoter allows some expression of MutH. In this condition, the distribution of foci lifetime is strikingly bimodal (in cyan in Fig. 3a). The high mode is similar to the mode of the foci lifetime distribution in the *mutH* strain and peaks at ~26 min, which corresponds to the doubling time in LB medium (26.6 min). The low mode is similar to the foci lifetime distribution in the WT strain and peaks at 2 min, corresponding to foci that are detected on a single frame. When arabinose is added to the medium, allowing an increase in MutH concentration and repair efficiency, the proportion of long-lived foci decreases, as shown by the gray distribution in Fig. 3a, where the height of the high mode substantially decreases in favor of an increase of the first mode at 2 min.

Modulating the MutH expression should change repair efficiency but not the total rate of error production. However, since the 2-min

time step for image acquisition in our experiments is larger than the average lifetime of short-lived foci, a substantial fraction of these foci is not detected, in contrast to long-lived ones. Thus, a change in the proportion of long-lived and short-lived foci will lead to a change in the rate of detected foci. This effect can be corrected based on the distribution of foci lifetimes (Supplementary Note 1 subsection on estimating the loss of YFP-MutL foci appearance due to 2-min discrete observations). We found, as expected, that the total error production rate does not change in the 4 conditions of Fig. 3a (Supplementary Fig. 4d), only the proportion of long-lived and short-lived foci changes (Supplementary Table 7).

The bimodality of the distribution in *mutH* pMutH strain growing in the absence of arabinose allowed us to determine a cut-off value at 14 min that allowed us to separate YFP-MutL foci into two distinct subpopulations (dashed line in Fig. 3a, Supplementary Fig. 5 and Supplementary Note 1 subsection on the method used to determine the cut-off value): short-lived and long-lived foci. Note that reasonable changes to this cut-off value do not impact the conclusions presented below (Supplementary Note 1 subsection on the variation of +/− 2 min in the cut-off value). These results demonstrate that successful and unsuccessful repair events can be detected and distinguished in a single bacterial strain, as short-lived and long-lived YFP-MutL foci.

To confirm that we can precisely quantify repaired and unrepaired replication errors, i.e., mutations, we estimated the mutation rate from the rate of long-lived YFP-MutL foci (Supplementary Note 1 subsection on estimating the mutation rate from YFP-MutL foci) and compared it to the mutation rate estimate obtained by MA + WGS. No significant differences (all *p*-values ≥ 0.09, Supplementary Table 8) could be

detected between the two mutation rate estimates, over a ~150-fold range of mutation rate (Fig. 3b). This conclusion is robust to changes in the cut-off value separating long and short-lived foci (Supplementary Note 1 subsection on the variation of +/−2 min in the cut-off value, Supplementary Fig. 6). Altogether, these results show that this imaging approach allows simultaneous detection of both repaired and unrepaired replication errors, and their precise quantification (Fig. 3c and Fig. 2).

## Error repair occurs on a timescale of 1 min

The demonstration that short-lived YFP-MutL foci in *E. coli* WT cells mark successful repair events opens the possibility of estimating for the first time the kinetics of error repair by methyl-directed MMR in vivo. This can be achieved by estimating the lifetime of these foci. In a successful repair event, errors are eliminated once MutH incises the new strand and UvrD unwinds the incised strand beyond the error. The unwound part of the new strand is then excised and resynthesized. The disappearance of the YFP-MutL focus marks the moment of error elimination, allowing us to track the repair reaction up to this crucial point using YFP-MutL foci.

To estimate the lifetime of short-lived foci, we first carried out several control experiments. Using short-term experiments (~4 min) and different imaging time steps leading to different bleaching rates (Supplementary Fig. 2a), we first verified that the lifetime distributions of YFP-MutL foci were not affected by photobleaching (Supplementary Fig. 2c). In our strain, YFP-MutL is under the control of the Plac promoter, which is stronger than the native MutL promoter (Pnat). We therefore verified that overexpression from Plac does not significantly change the foci lifetimes (Fig. 1e and Supplementary Fig. 1). Finally, we find that the average YFP-MutL focus lifetime is ~40 s (Supplementary Note 1 subsection on estimating error repair kinetics), which gives an estimation of the timescale of faithful repair by MMR in *E. coli*.

## Many spontaneous mutations result from errors that are detected but inefficiently repaired

Uncorrected replication errors (1%) are a major cause of spontaneous mutations observed in actively dividing *E. coli* cells[2], but the reason why repair fails for some errors remains unknown. The efficiency of error detection could be limited. Alternatively, the repair could fail on detected errors. Quantification of long-lived foci in the WT strain from long-term time-lapse experiments shows that the rate of these foci (0.0014/generation) matches the mutation rate obtained by Lee et al. using MA + WGS[2] (Fig. 3b). Likewise, in agreement with classical genetic studies that estimated that MMR fails to repair ~1% of the errors, we find that 1% of foci are long-lived in WT cells (Supplementary Table 7). Since errors that escape MMR detection would also escape our detection by YFP-MutL, this demonstrates that a substantial fraction of mutations are caused by MMR failures on detected errors (Fig. 2). These results demonstrate the relevance of this approach in studying MMR efficiency at the single-cell level in (WT) *E. coli*.

## Limited repair efficiency is due to a temporal constraint imposed by the strand discrimination signal

We next investigated the molecular mechanisms underlying repair failures. The effectiveness of repair relies on the cleavage of a hemimethylated GATC site near the error by MutH. This cleavage has to occur before the GATC site is methylated by Dam. We thus hypothesized that Dam outcompeting MutH for GATC binding could account for the failures observed during the MMR repair step. To test this hypothesis, we manipulated the levels of Dam and MutH in *E. coli* cells by overproduction or depletion and assessed the rate of long-lived foci. As shown in Fig. 3a, b and discussed above, decreasing the level of MutH leads to an increase in the rate of long-lived foci, and a bimodal distribution of foci lifetime, representing the two populations of repaired and unrepaired errors. Likewise, in agreement with our

hypothesis, and consistent with the increased mutation rate of Dam overproducing cells[7], a strain carrying an additional chromosomal copy of the *dam* gene (WT PLtetO1-*dam*) exhibits an increased rate of long-lived foci compared to WT cells (Fig. 4a). This rate is smaller than the rate of MMR-deficient *mutH* cells (Fig. 4a), suggesting a partial defect in MMR. Note that in contrast to the situation where MutH level is modulated (in *mutH* pMutH), the distribution of foci lifetime cannot be interpreted easily when the level of Dam is increased (Fig. 4b and Supplementary Note 1 on YFP-MutL foci in cells with varying levels of Dam and MutH), since Dam is involved in the control of replication initiation. In a strain overexpressing Dam, the inter-initiation period is much more variable[22], unrepaired errors can therefore lead to fluorescent foci of shorter or longer lifetimes than in WT.

Conversely, both *dam* deletion and overproduction of MutH (WT pMutH strain) resulted in a decreased rate of long-lived foci compared to WT cells (≥2-fold, Fig. 4a and Supplementary Table 9), indicating enhanced repair efficiency. This suggests that MutH is a limiting factor in MMR and this limitation accounts for at least 50% of repair failures. Overall, our results demonstrate that in *E. coli*, the fate of replication errors is determined by the competition between two reactions: one initiates the repair process, while the other governs the duration of the repair time window (Fig. 4c).

## Repair failures are not due to MMR saturation and are not associated with aged cells

Having established repair failures as an important source of spontaneous mutations that can be detected with YFP-MutL foci, we next investigated whether these failures occur in cells with specific phenotypes. We first investigated whether cells that contain long-lived foci have an increased total rate of error production compared to the rest of the population, as previous studies showed that repair efficiency can decrease when the rate of replication errors increases above a certain threshold, in polymerase mutants or in presence of mutagenic agents[23-26]. We found no significant difference (Welch's *t*-test *p*-value = 0.69) in the total rate of YFP-MutL foci in cells that have a long-lived focus, compared to the other cells (Fig. 5a and Supplementary Note 1). This shows that repair failures are not due to saturation of the repair system.

In the mother machine, mother cells are retained at the end of the microchannels, leading to a progressive increase in their replicative age with each division[27]. We investigated whether the repair efficiency and error production change with age, over the course of 100 generations. The rate of all YFP-MutL foci does not change significantly (Supplementary Fig. 10), indicating a stable error production. Likewise, aging does not impact repair efficiency, as evidenced by the stability of the ratio between the rate of long-lived YFP-MutL foci and the rate of all YFP-MutL foci (Fig. 5b, Supplementary Note 1 subsection on repair efficiency over replicative age).

## Repair efficiency is variable among isogenic cells, leading to non-Poissonian dynamics of spontaneous mutations

Methods to estimate mutation rates typically rely on the assumption that every cell in the population has an equal probability of undergoing mutation. However, repair efficiency may fluctuate through time and among cells due, for example, to fluctuating concentrations of repair enzymes. Replication errors, a major source of mutations in growing cells, occur according to a Poisson process, as we previously demonstrated[17], and they can either be repaired successfully or not. Repair is therefore like a coin toss experiment, i.e., in mathematical terms a Bernoulli trial. If repair efficiency is homogeneous between cells, i.e., the probability of success of the Bernoulli trial is constant, the resulting mutation dynamics is a compound Poisson process with Bernoulli increments, i.e., a Poisson process with a rate that is the product of the rate of replication errors by the probability of repair failure. Therefore, Poissonian error production combined with

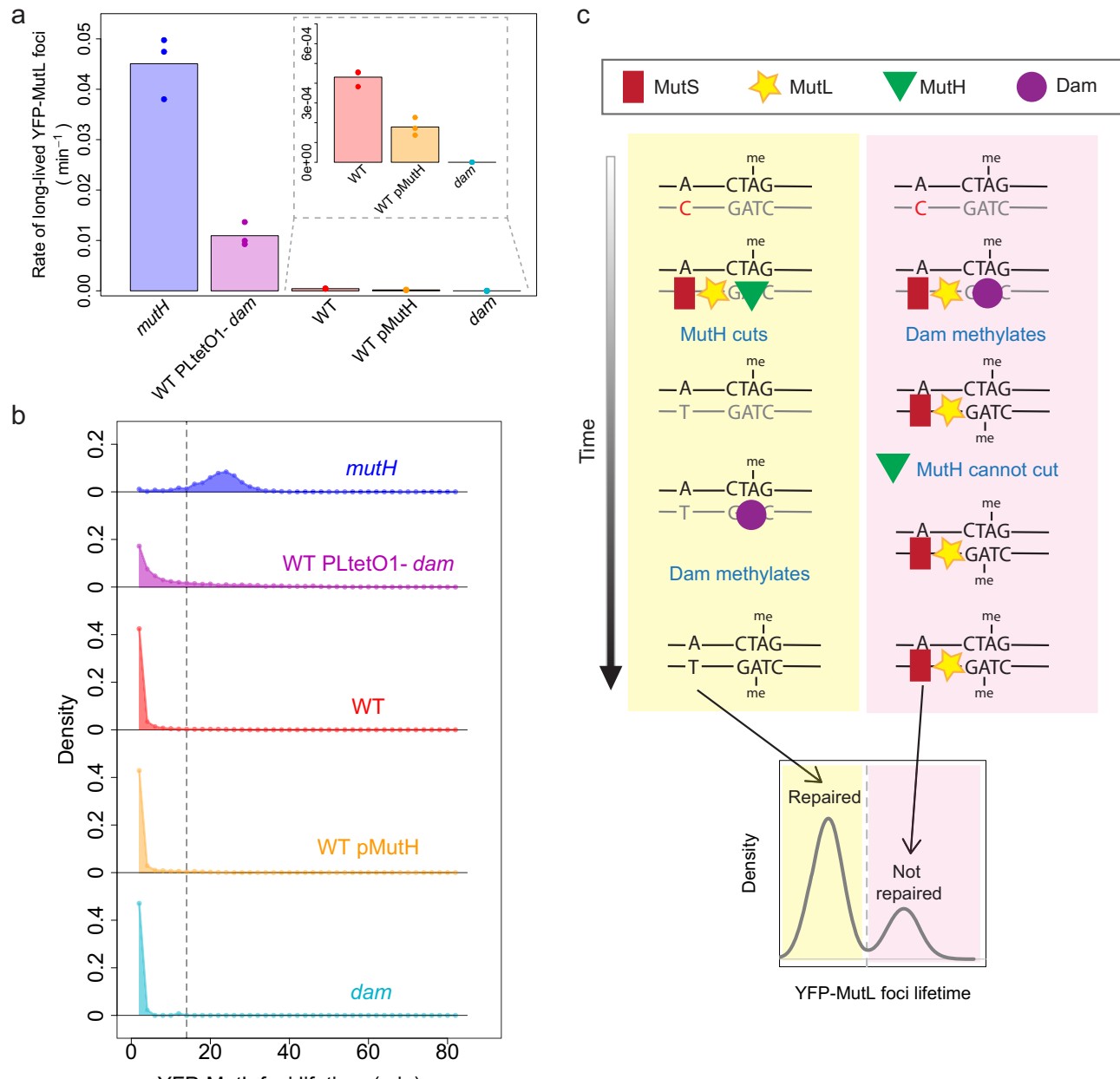

**Fig. 4 | Competition between Dam-mediated DNA methylation and strand cleavage by MutH causes MMR failures.** Repair efficiency is modulated by deletion of *mutH* (*mutH*, blue), overexpression of Dam (WT PLtetO1-*dam*, violet), overexpression of MutH (WT pMutH, yellow), and deletion of *dam* (*dam*, cyan). These conditions are compared to the wild-type (WT, red). **a** Rate of long-lived foci. (inset: zoom-in of the results for WT, WT pMutH, and *dam*). The bars represent the average of individual experiments (dots, 3 experiments per condition except for *dam* for which there are 4 experiments). **b** The distribution of YFP-MutL foci lifetimes is shown for one representative experiment of each condition shown in panel

(**a**). All replicates are presented in Supplementary Fig. 7. **c** Competition between Dam and MutH determines the fate of replication errors detected by MMR. After MutS (red rectangle) and MutL (yellow star) binding, if MutH (green triangle) binds to the unmethylated GATC site before Dam (purple circle), the error is repaired, leading to a short-lived YFP-MutL focus (left side, yellow-shaded areas of the schematic). On the contrary, if Dam binds earlier than MutH to the unmethylated GATC site, MutH cannot initiate repair, leading to a long-lived YFP-MutL focus representative of a nascent mutation (right side, pink-shaded areas of the schematic). Source data are provided as a Source Data file.

homogeneous repair capacity should lead to a Poissonian dynamics of mutations. In contrast, non-Poissonian dynamics of mutations would indicate fluctuations in repair efficiency through time and among cells. To investigate variability in repair efficiency, we tracked ~ 20,000 YFP-MutL foci, ~500 of which were long-lived, occurring in mother cell lineages that we followed for ~100 consecutive generations and we calculated their inter-arrival times (Fig. 6a, b). Events that occur randomly at a constant rate, i.e., following a Poisson dynamics, should result in exponentially distributed inter-arrival times, as we showed

previously for replication error production[17] using a *mutH* strain (Fig. 6c). Here we follow specifically the errors that MMR fails to repair, i.e., nascent mutations, detected as long-lived YFP-MutL foci in WT cells. We find that their inter-arrival times are not exponentially distributed, therefore showing that mutation dynamics is not Poissonian (Fig. 6d, Supplementary Note 1 on correcting for bias in the distribution of inter-arrival times of long-lived foci, and Supplementary Figs. 11 and 12). Note that this conclusion is robust to changes in the cut-off value defining long-lived foci (Supplementary Note 1

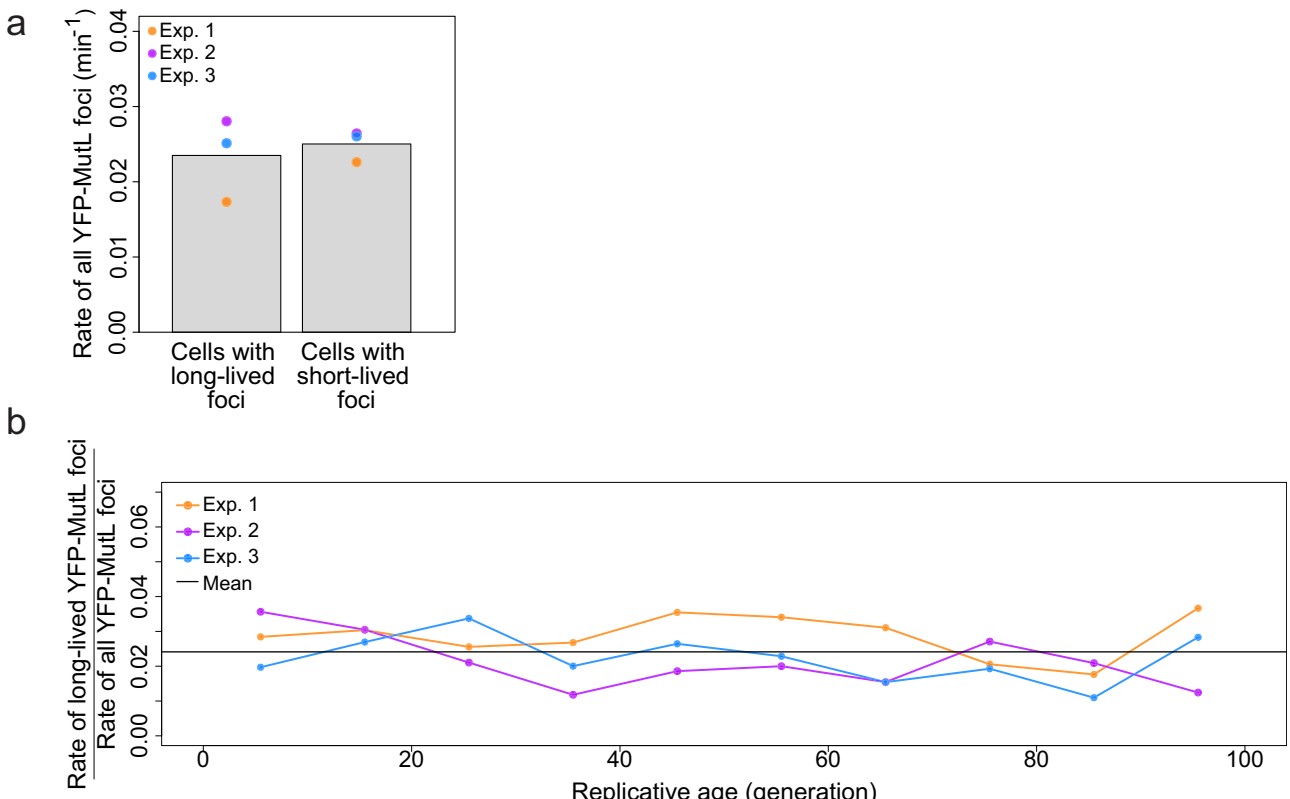

**Fig. 5 | Repair failures are not due to increased production of replication errors and do not increase with replicative age. a** Comparison of the rate of all YFP-MutL foci in (i) cells showing a long-lived focus and (ii) cells showing a short-lived focus (see dedicated subsection in Supplementary Note 1). The bars represent the average of three experiments (colored dots), with non-significant differences (two-sided Welch's *t*-test, *p*-value = 0.69). **b** Repair efficiency, estimated as the ratio of the rate of long-lived YFP-MutL foci to the rate of all YFP-MutL foci, is plotted against the cells' replicative age. Each color represents a different experiment. Dots are obtained by binning the cells by 10 generations. The horizontal black line is the average of all the data. No significant difference between the first and the last 10 generations (Barnard's unconditional test, two-sided *p*-value = 0.64, Supplementary Table 10 and dedicated subsection in Supplementary Note 1). Source data are provided as a Source Data file.

subsection on the variation of +/− 2 min in the cut-off value and Supplementary Fig. 13). This demonstrates that the efficiency of error repair is heterogeneous among isogenic cells. More precisely, Fig. 6d shows that short inter-arrival times between mutations (<40 min) are over-represented, indicating the presence of transient states of decreased repair efficiency and therefore elevated mutation rate, lasting a few generations (typically <1–2 generations, affecting single lineages, i.e., single microchannels, independently).

## Discussion

Replication errors are a major source of spontaneous mutations in growing cells[1,2]. But it is the MMR, a system dedicated to error repair, that determines whether the error will give rise to a mutation. This conserved repair process has been studied for over forty years[28]. However, in the absence of appropriate tools, it has so far been impossible to characterize its efficiency at the level of individual cells, which has prevented the characterization of the dynamics of repair failures and therefore of spontaneous mutations in genomes. Former studies have exploited foci of fluorescently labeled MutL or homolog proteins to quantify replication errors, but whether the detected foci corresponded to all errors[17,20], only repaired errors[29], or even only the 1% of unrepaired errors[12,19] remained unknown. Using a combination of YFP-MutL, time-lapse microscopy, and microfluidics, we show here that the detection of YFP-MutL foci and quantification of their lifetimes allow us to detect all errors in repair-proficient cells and distinguish repaired from unrepaired ones and thus to study the efficiency of error correction in individual cells.

This approach allowed quantifying the lifetimes of all corrected errors, thus estimating the kinetics of the error repair reaction in vivo. We found that error repair takes typically less than a minute. Previous studies of *E. coli* MMR kinetics have been carried out in vitro, yielding highly variable results. For instance, while studies based on *E. coli* cell extracts report a typical timescale of 1 h[30,31], others using purified proteins indicate 1 min, in agreement with our results[32,33]. Likewise, for non-methyl-directed MMR, repair time has been estimated to be 1.5 min employing fluorescently tagged MutL homologs in *S. cerevisiae*[29].

We investigated the origin of mutations. Although most spontaneous mutations originate from replication errors in growing cells, it is not known why some are not repaired in cells with functional MMR. For instance, error detection could be ineffective, repair could fail, or both. We found that 1% of YFP-MutL foci are long-lived, i.e., 1% of errors that are detected by MMR are not successfully repaired. Since this proportion matches the proportion of errors that lead to a mutation, this suggests that a substantial fraction of spontaneous mutations are caused by MMR failures at the repair stage. This is in agreement with previous genetic studies, showing that MutS overexpression has no impact on mutation rate[34–36]. However, the methods used in these studies could hardly detect any <50% change in mutation rate. Likewise, our method cannot be used to precisely quantify MMR detection efficiency, since 99% of errors are repaired and overexpression of MutS would therefore lead to a <1% increase in the number of YFP-MutL foci, which is too low to be reliably detected. However, although it is likely that a fraction of spontaneous mutations is due to inefficient error

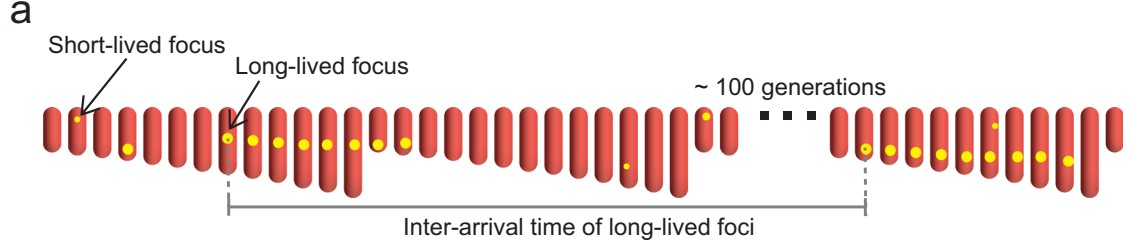

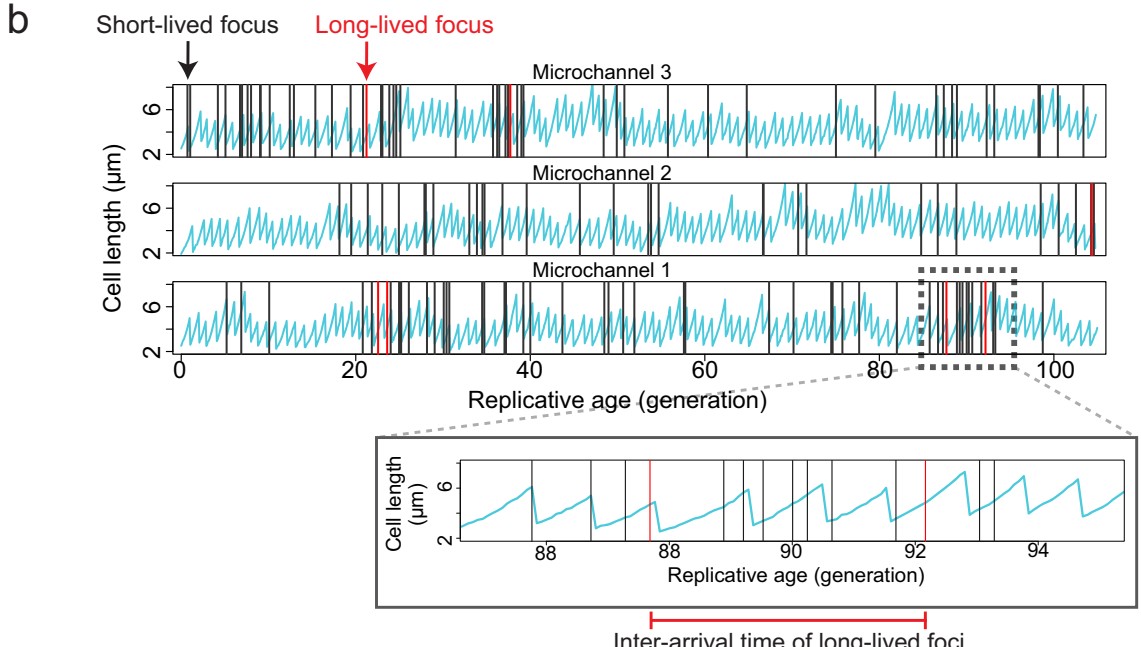

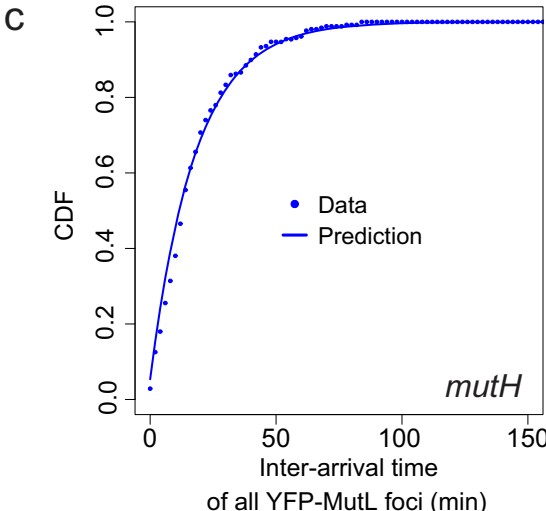

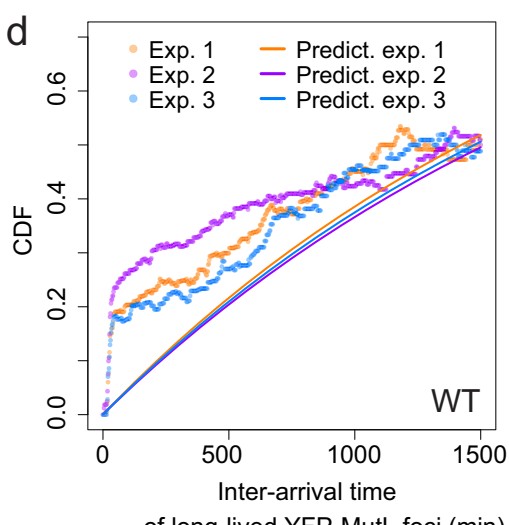

detection, our findings demonstrate that failures to repair detected errors explain many mutations. Our work also gives insights into the molecular causes of such repair failures. After replication, the newly synthesized strand remains unmethylated for several minutes in *E. coli*[8,9]. Since MutH cannot cut fully methylated DNA, error repair can only function effectively during this short time window. We show that

prevention of methylation and overproduction of MutH lead to improved repair efficiency. This suggests that a competition between the repair initiation reaction and the reaction determining the duration of the repair time window determines the fate of replication errors in *E. coli*. Previous genetic studies examined whether MMR proteins other than MutS were limiting but the results reported vary from study to

**Fig. 6 | Non-Poissonian occurrence of spontaneous mutations. a** The variable analyzed is the inter-arrival time between two successive YFP-MutL foci (yellow dots inside cells depicted in red). **b** Three representative examples of mother cell length over 100 replications (Microchannels 1, 2, 3). The vertical lines represent the times of occurrence of short-lived (black) and long-lived (red) YFP-MutL foci. The zoom-in of Microchannel 1 shows an example of inter-arrival time between two successive long-lived foci. **c** Cumulative distribution function (CDF) of the inter-arrival times for all YFP-MutL foci detected in *mutH* cells. The dots represent the data from one representative experiment, the solid line depicts the analytical

prediction for a Poisson process (i.e., the CDF of an exponential distribution corrected for discrete observations, Supplementary Note 1 subsection on bias correction) with a rate given by the empirical rate of occurrence of all YFP-MutL foci. **d** Analog of panel (**c**) showing the results for long-lived YFP-MutL foci in three WT experiments (each color represents one experiment). The dots represent empirical data, solid lines depict the analytical prediction for a Poisson process. Each experiment has its prediction calculated by using its empirical rate (Supplementary Note 1 subsection on bias correction and Supplementary Figs. 11–13). Source data are provided as a Source Data file.

study[34–36]. These discrepancies may be attributed to the limitations of the experimental systems used, which were inadequate for detecting small differences and thus addressing these questions effectively.

Although the MMR system is highly conserved across various organisms, the majority of prokaryotic and eukaryotic species do not rely on DNA methylation to guide repair to the newly synthesized DNA strand. In these organisms, the strand discrimination signal has not been conclusively identified but it is commonly assumed to involve replication-associated strand discontinuities and nicks. In *S. cerevisiae*, a model organism for non-methyl-directed MMR, a repair window of approximately 10 to 15 min has been reported[37]. This observation suggests a time constraint for error correction, similar to what is observed for methyl-directed MMR. Studies in *S. cerevisiae* have additionally demonstrated that overexpression of DNA ligase I (Cdc9), causing premature ligation of replication-associated nicks, leads to an elevated mutation rate and the accumulation of fluorescent foci of the MutL homolog, Pms1[38]. Conversely, reducing or delaying Cdc9 expression prolongs the temporal window in which loci can undergo MMR[38]. This indicates that, as reported here for methyl-directed MMR, the effectiveness of non-methyl-directed MMR can be modified by modulating the temporal window available for repair. In light of our results, this suggests that an evolutionarily conserved time constraint on MMR might be setting a limit on its efficiency and explaining the origin of an important part of spontaneous mutations universally.

Finally, this work describes the temporal dynamics of spontaneous mutations by tracking error repair failures over time in thousands of single cells. Conventional methods to estimate mutation rate, including the famous fluctuation test (Luria-Delbrück test), assume that the rate is homogeneous within a population of cell[39,40]. However, theoretical and experimental evidence suggests that isogenic populations may contain subpopulations of cells with transiently higher mutation rates[12–14]. Here we demonstrate the existence of such subpopulations in unstressed, growing bacteria, and show that they are due to cell-to-cell variability in repair capacity. Variability in mutation rate such as demonstrated here may have an impact on the fitness of the population at mutation-selection balance and could play a role in adaptation, by facilitating the acquisition of combinations of mutation[15]. It may therefore have an impact on the emergence of antibiotic resistance[41].

## Methods
### Microfluidic experiments
The mother machine device is described in Robert et al.[17]. It consists of a main trench 15 mm long, 50 μm wide, and ~30 μm high and ~1000 perpendicular microchannels which are ~25 μm long and ~1 μm wide and high. To produce PDMS chips from the molds we followed the protocol indicated in Robert et al.[17,18]. For delivering and controlling the flow of growth medium into the chip, we used a Harvard Apparatus PHD Ultra syringe pump set at 2 ml/h, 50 ml Monoject syringes with LS23 needles (Phymep), Tygon S54-HL flexible tubing (Phymep) and SC23/8 steel couplers (Phymep). All experiments were performed with LB as a growth medium. To induce *yfp-mutL* expression, the medium was supplemented with 1 mM IPTG. To select for the plasmid pBAD24MutH ampicillin (50 μg/ml) was also added to the growth medium. For experiments where we modulated the cellular level of

MutH (pBAD24MutH) and DnaQ926, the growth medium was changed during the experiment from LB to LB supplemented with arabinose 0.05%, 0.1% or 0.5% (as indicated in the text).

### Microscopy
Fluorescence imaging was performed as in Robert et al.[17] with a Delta Vision Elite inverted microscope equipped with the Ultimate Focus system for automatic focalization, a 100x oil immersion objective (N.A. 1.4), a temperature-controlled chamber, and the DV Elite sCMOS camera. Fluorescence illumination was provided by the DV Light Solid-State Illuminator 7 Colors at 575 nm (mCherry excitation) and 513 nm (YFP excitation). The temperature controller was set to 37 °C. The waiting time before acquisition was 3 h to allow the cells to adapt to the environment and reach stable exponential growth. On average, 20 fields of view were selected per experiment, with an average of ~20 microchannels per field of view. For short-term experiments, images were acquired every $\Delta t = 7.5$ s, 15 s, or 30 s for ~4 min. For long-term experiments, images were acquired every $\Delta t = 2$ min for ~11 to 46 h. For mCherry illumination, we used an exposure time of 0.2 s and a maximum LED intensity of 3%, and for YFP illumination, an exposure time of 2 s and a maximum LED intensity of 7%. As in Robert et al.[17], for YFP images, we used the optical axis integration imaging mode, which allows collecting fluorescent light by integrating one image through a z-axis movement of 1 μm around the cell focal plane. All experiments were repeated at least twice.

### Image analysis
Images were analyzed using BACMMAN, a custom ImageJ plugin[16,21]. Segmentation and tracking of the microchannels over time were performed using mCherry images and the *MicrochannelFluo2D* and *MicrochannelTracker* modules. Bacteria segmentation and tracking from mCherry images were performed using the *BacteriaFluo* and *BacteriaClosedMicrochannelTrackerLocalCorrections* modules. YFP-MutL foci were segmented and tracked over time from YFP images using the *SpotDetector*, *SpotSegmenter*, and *NestedSpotTracker* modules. BACMMAN automatically measures characteristics of bacteria (e.g., growth rate, cell size, and length) and foci (e.g., lifetime and time of onset).

### Data analysis
Data analysis was performed using custom codes developed in R language. In the case of long-term experiments, analysis was limited to mother cells. To improve statistical power, for the study of YFP-MutL foci rates and lifetimes in short-term experiments, and the analysis of the bleaching, all cells (mothers and daughters) were taken into account (first, second, and fourth subsections of Supplementary Note 1). Simulations were performed with custom scripts in R and *Wolfram Mathematica* languages. All the details on data analysis are provided in Supplementary Note 1.

### Mutation accumulation + whole genome sequencing experiment (MA + WGS)
MA lines were propagated at 37 °C in an LB liquid medium containing 1 mM IPTG to induce *yfp-mutL* expression. Additionally, ampicillin (50 μg/ml) was included in the growth medium to select for

pBAD24MutH, and arabinose at concentrations of 0.05-0.5% was used to induce the expression of MutH from pBAD24MutH or DnaQ926 in the MF1R and MVEC253 strains, as specified in the manuscript text and/or figure legends. At least 4 MA lines were propagated for each strain and condition by diluting 5*10^8-fold every day, over 2 or 4 days, depending on the mutation rate of the strain. See the first subsection of Supplementary Note 2 for more details. To estimate the number of generations in each cycle, appropriate dilutions of the saturated cultures were spread on LB plates. Details can be found in the second subsection of Supplementary Note 2. Genomic DNA was extracted from 1 ml of frozen cultures using the Wizard® Genomic DNA Purification Kit (Promega) following the supplier's instructions, with the exception of DNA rehydration, which was performed in 10 mM Tris pH 8. The DNA samples were quantified using a Qubit 4 fluorometer (Invitrogen) and the Qubit dsDNA BR Assay Kit, following the supplier's instructions. For sequencing, 500 ng of DNA samples were sent to Eurofins Genomics (Eurofins, France). The sequencing was performed on an Illumina platform using 2 ×150 bp paired-end read mode, with the NovaSeq 6000 S4 PE150 XP instrument, following standard procedures. The average coverage was approximately 5 million read pairs for each MA line (at least 3.75 million read pairs), providing at least 100x average coverage. We used an open-source computational pipeline called breseq (versions 0.36.0)[42] to predict mutations in the samples based on how Illumina reads align to each position in the *E. coli* MG1655 genome (GenBank: NC_000913). Details can be found in Supplementary Note 2.

## Reporting summary

Further information on research design is available in the Nature Portfolio Reporting Summary linked to this article.

## Data availability

The image analysis measurements, NGS, and breseq data generated in this study have been deposited in the "Data for Real-time monitoring of replication errors" database at https://doi.org/10.57745/UFHTYU. The reference genome sequence was GenBank NC_000913. Source data are provided in this paper.

## Code availability

Codes used for the main results of the manuscript are publicly available at https://doi.org/10.57745/UFHTYU.

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

## Acknowledgements

This work was supported by the city of Paris through the Emergences 2018 program (to M.E.), by the Agence Nationale de Recherche grant ANR-19-CE35-0003 (TRANSMUTATOR) (to L.R.), and by Impulscience of the Fondation Bettencourt-Schueller (to L.R.). The authors thank Romain Koszul for discussions and comments on the manuscript and Maria Giralt-Zuñiga and Alexandre Deloupy for technical help with strain constructions.

## Author contributions

Conceptualization: M.E., L.R. Investigation: M.E., C.E.B., J.O., M.V., M.D.P. Analysis: M.E., L.R., C.E.B. Supervision: M.E., L.R. Writing: M.E., L.R., C.E.B.

## Competing interests

The authors declare no competing interests.
