## [Peer Review File · Nature Communications]

Real-time monitoring of replication errors' fate reveals the origin and dynamics of spontaneous mutationsReviewer #1 (Remarks to the Author):

Bene et al. use microfluidic fluorescence in a mother machine (cells grow out from a single mother cell) to determine the nature of MMR failures. They track YFP-MutL using fluorescent imaging to determine repair efficiency and kinetics.

The work is novel and interesting, it allows for detection of enzyme kinetics of repair in *E. coli* and uses advanced microfluidic technology to do so expanding upon their previous work.

The authors do considerable work in examining mutational process and the conclusions regarding the non-poissonian dynamics and repair kinetics is significant to the field.

Overall, the mutL kinetics are convincing as the authors see similar YFP-mutL occurrences between mutH, WT, and MF1R, furthermore they see a longer illumination in mutH strains supporting the data. Although, there is rough bleaching I don't think that can be avoided and the authors describe and correct for the issues well. The methodology overall is sound - desc of MA needs clarification. Fig 4B has some issues.

Overall, this is a strong paper, generally well written but needs to expand on the setup, and covers many interesting topics. Some topics need to be further elaborated on or described to reach their conclusions.

Major

1) A better description of the mother machine is needed in the introduction or first part of results. At least one paragraph describing the setup, the average number of cell divisions in a channel, average number of YFP events in the channel. Channels per experiment, experiments run. That *ecoli* was the organism used in the system – prior to the fourth paragraph in the results. The first part of the results section is very vague and requires significant knowledge about the previous work. This needs to be improved.

2) There are some issues with the comparisons in Fig 4B. Although it is ok to state the rate of YFP-mutL over time (Supp.text 2.11) with significant corrections (2.1.5-2.2.17), it doesn't work when you are comparing two different cells with different segmentation times, e.g., filamentous and normal length. For example, 4 filamentous cells would grow over a certain time interval vs 3 normal length. Furthermore, it is unlikely that the filamentous cells would be all within a single time unit to be accurately measured over a comparable time scale. Finally, bleaching – the effects which are vague and hide YFP-events would have less effect on filamentous vs normal length. There are some issues with the comparison made in Fig 4B that need to be clarified, corrected for, or removed.

Minor

The authors describe the use of a classical mutation accumulation but they use serial dilution (Supp. Text 2.2.1). This is not a classical approach of colony transfer as done by Lee et al., and allows for some inclusion of selection on mutations as this increases the accumulation population from roughly 1000 (cells in a colony) to 50000++ (diluted cells) – allowing for purification of some more deleterious mutations through selection. This should be noted in the text and whether it effects Fig. 2B. This is minor as the MA+WGS was just used for confirmation purposes.

Reviewer #2 (Remarks to the Author):

In this work, the authors expanded on their previous work in which they visualized point mutations in

Escherichia coli using a fluorescently-labeled component of the mutHLS mismatch repair (MMR) system, mutL. Using a mother machine microfluidic device and fluorescence microscopy, they were able to visualize the occurrence of fluorescent foci within growing cells that correspond to instances of errors in DNA replication. In the present work, the authors are able to distinguish between repaired and unrepaired errors in DNA replication by examining the persistence time of the corresponding fluorescent foci, thereby providing concrete, *in vivo* observations about the kinetics of methylation-directed mismatch repair in *E. coli* and the origin of single nucleotide mutations.

The authors show that whether or not a given replication error gets repaired, is determined by the competition between two processes: the rate of error recognition by the MMR complex and the rate of methylation by the dam methylase. The experimental system developed by the authors is able to simultaneously track the occurrence of repaired and unrepaired errors in DNA replication and make good estimates of single-base mutation rates. Furthermore, they are able to detect subpopulations of cells with diminished repair capacity, as distinguished by their non-Poissonian mutation dynamics.

Major points:

1. It would help to have a cartoon that depicts the logic of the assay the authors are using. If one is away from the field, it takes quite some work through the text to make oneself an image of the elegance of the assay.
2. The authors state that the inter-arrival times of long-lived foci in "mother cells", which represent unrepaired errors, are not exponentially distributed. This indicates that the error production process is no longer Poissonian, which is compelling in light of the authors' observations that there is no effect on repair efficiency in aging cells. In figure 5C, three examples of this non-Poissonian behavior are shown. Of these experiments, the green trace (experiment 3) seems to display a strikingly distinct behavior from the other two. In context of the data presented in figure 4C, where the authors show the aging-independence of the repair efficiency, the data in figure 5C seem incomplete. It would help that the authors add two plots to figure 5, showing the absolute time-trace over replicative age (as opposed to the cumulative distribution function) of both the short-lived foci arrivals as well as specifically the long-lived foci arrivals. This would be valuable information for any reader.

Furthermore, it is not clear from the text whether the non-random mutagenesis is evident in the whole population at different times, or whether this phenotype is only evident in a small subpopulation of the growing cells. While this phenomenon is compelling enough to warrant a dedicated investigation, I recognize that additional experiments in this direction are outside the scope of this work. However, for this work it is critical for the authors to clarify this observation in the text, as it goes directly to their conclusion that the origin of single nucleotide mutations is driven by the rate competition between repair efficiency and methylation. It would be helpful to illustrate this observation with a cartoon.

3. Regarding the seventh subsection of the Results section (Repair failures are not due to...): the authors state that the filamentous phenotype they observe indicates the induction of some kind of stress response within the cell (e.g. SOS response), but that they do not observe a change in the error repair rate between filamentous and non-filamentous cells. However, a primary result from the study of stress-induced mutagenesis¹ is that two of the primary stress responses of *E. coli*, namely the SOS response and rpoS-mediated stress response, are associated with both the up-regulation of error-prone polymerases, polIV and polV, as well as the down-regulation of MMR machinery. I would expect to see an increase in the number of unrepaired errors, or at the very least in the number of total foci in these cells. I am surprised that the authors observe no difference at all in the filamentous cells. The authors should include the data about total foci observed in the filamentous cells, presented in comparison to those observed in the unstressed cells. Furthermore, it would be quite valuable if the authors were able to further clarify the stress-response phenotype of the filamentous cells, as it is currently unclear in the text.

The link between stress responses and single base pair mutations represents an interesting research direction. Two possible further experiments would be to make the same observations of short and long-lived foci in a strain of *E. coli* that has an inducible SOS response, and in wild-type cells growing in stressful conditions. However, these would be outside the scope of the paper as it stands.

4. Is there a bias in the spectrum of errors that are repaired versus those that are unrepaired? As the authors have determined that error repair is governed by the physico-chemical constraints of the rates of error detection and DNA methylation, understanding whether and how these constraints impose bias on the types of errors that are repaired would be valuable for understanding the repair kinetics. Such analysis could be achieved simply by examining data from the whole genome analyses for both WT and repair-deficient strains without any further experiments.

Minor Points

i. At the end of the third paragraph of the Discussion section, the authors indicate that their results suggest an evolutionarily conserved time constraint on the action of mismatch repair systems that may be universally significant. This observation has important consequences for our understanding of the origins of mutations, but is only treated superficially here. It would be a valuable addition to this paper for the authors to support this observation with a longer, literature-based discussion about the evolutionary context of MMR systems and the methylation machinery.

ii. In figures 2A and 3B, the arrangement of the distributions of foci lifetimes along a diagonal "z"-axis is confusing and detracts from the otherwise compelling point made by these figures. I suggest that these plots be rearranged into a two-dimensional figure so that the "x"-axes of the distributions are aligned with one another.

iii. In the second paragraph of the third Results subsection, the authors state that the observed average lifetime of the fluorescent foci is forty seconds, which corresponds to the timescale of the MMR repair activity. However, it is not clear from the text if this average time includes the minority of long-lived foci, which correspond to unrepaired errors.

iv. Please check that throughout the text the Fig numbers correspond to the actual figures, for example on page 7 last paragraph Figure S10 should be S11.

Overall, this is an elegant experimental work, that sheds light on the process of mismatch repair.

Citations:

1) Foster, P. L. (2007). Stress-induced mutagenesis in bacteria. *Critical Reviews in Biochemistry and Molecular Biology*, 42(5), 373–397. <https://doi.org/10.1080/10409230701648494>

Reviewer #3 (Remarks to the Author):

In this manuscript, the authors combined microfluidics and time-lapse microscopy to address the fundamental question regarding the repair efficiency of MMR within *E. coli*. Overall, the control experiments were well-designed and validated their methodology. By monitoring in real-time the fate of replication errors, the authors unveiled the stochasticity in MMR efficiency within the isogenic population, which could be an important source of spontaneous mutations. I would recommend the acceptance of the manuscript after the authors address my questions below.

Major questions:

1. The description related to the data and conclusion relate to Fig. 5 is unclear. First, does the repair efficiency change over time in the same cell? If the three experiments are from one cell per experiment, I will agree with the authors' conclusion. If it comes from multiple cells, I would expect to see similar curves as the population average should be similar. please make this clear before drawing the conclusion "the efficiency of error repair is heterogeneous among isogenic cells".

2. Also the initial jump in Fig. 5C is almost similar to the trend observed in the mutH cells in Fig. 5B, seems to me that there are two processes going on here, one is searching process and one is the repairing process. Based on Fig.5B, I will think that the searching process is likely Poissonian. Then the repairing process is likely non-Poissonian as the authors were measuring the time intervals between the two foci, which depends on the arrival time of the second foci as well as the repairing time in the first foci. In fact, I will not expect it will follow the Poisson process. I would strongly recommend the authors put on a simulation to recapitulate the dynamics in silico. Directly using the non-Poissonian behavior to conclude that "the error repair is heterogeneous" is not convincing.

Minor suggestions:

3. In Fig. S2A, the authors presented a control experiment to rule out the potential effects caused by photobleaching. The photobleaching of three conditions (7.5s, 15s, 30s) were shown in this plot. It would be better if the authors could include the longer-time conditions (e.g., 2min) used in this manuscript.

4. Please provide a summary of some basic metrics regarding the foci detection. For example, how many foci can be detected in each cell on average? How many short-lived foci and long-lived foci were detected in each single cell?

5. Regarding the long-term monitoring of MutL foci experiment (Fig. 5), it will be helpful to include a more detailed experimental scheme, like Fig. S12A, in the main figures.

We would like to thank the three reviewers for their valuable comments and suggestions. As indicated in the point-by-point response that follows, we have addressed all their concerns. This involves providing important clarifications, introducing significant changes to the manuscript text and figures and incorporating illustrations to facilitate the understanding of our experimental approach and key findings (Figures 2-5, S2 & S8 of the original manuscript are modified; new figures are added to the revised manuscript, Figure 2 & S12). We have also carried out numerical simulations (Figure S12) to further confirm the validity of our conclusions. Changes to the text are highlighted in blue for ease of access.

Reviewer 1

Question 1: A better description of the mother machine is needed in the introduction or first part of results. At least one paragraph describing the setup, the average number of cell divisions in a channel, average number of YFP events in the channel. Channels per experiment, experiments run. That *E. coli* was the organism used in the system – prior to the fourth paragraph in the results. The first part of the results section is very vague and requires significant knowledge about the previous work. This needs to be improved.

Answer 1: We agree with this comment and we modified the manuscript accordingly. Specifically, we included a description of our setup in the introduction, in the last paragraph. This description comprises key experimental details, such as the time step and duration of the experiments, the number of cell divisions observed in a single microchannel, as well as the typical number of microchannels imaged and fluorescent foci detected. In addition, we provided a more detailed description of our previous work in the second-to-last paragraph, to enhance the understanding of the initial part of the results section. For clarity, the last paragraph of introduction now states explicitly that *E. coli* is the organism used in this study (line 78)

Question 2: There are some issues with the comparisons in Fig 4B. Although it is ok to state the rate of YFP-mutL over time (Supp.text 2.11) with significant corrections (2.1.5-2.2.17), it doesn't work when you are comparing two different cells with different segmentation times, e.g.,

filamentous and normal length. For example, 4 filamentous cells would grow over a certain time interval vs 3 normal length. Furthermore, it is unlikely that the filamentous cells would be all within a single time unit to be accurately measured over a comparable time scale. Finally, bleaching – the effects which are vague and hide YFP-events would have less effect on filamentous vs normal length. There are some issues with the comparison made in Fig 4B that need to be clarified, corrected for, or removed.

Answer 2: Reviewer 2 also had some concerns about the result shown in Figure 4B. As we have elaborated in our response to his question below, we agree with reviewer 2 that clarification of this result is beyond the scope of our work. As suggested by Reviewer 1, we have therefore removed this analysis from the revised manuscript and modified the main text and supplementary information accordingly.

Minor Points

i. The authors describe the use of a classical mutation accumulation but they use serial dilution (Supp. Text 2.2.1). This is not a classical approach of colony transfer as done by Lee et al., and allows for some inclusion of selection on mutations as this increases the accumulation population from roughly 1000 (cells in a colony) to 50000++ (diluted cells) – allowing for purification of some more deleterious mutations through selection. This should be noted in the text and whether it effects Fig. 2B. This is minor as the MA+WGS was just used for confirmation purposes.

- We agree with the reviewer that the MA protocol we employed, using liquid cultures allows in principle for more selection than the classical one on solid medium. Note, however, that our approach involves strong dilutions, leading to a very small initial number of cells per culture (~10). In addition, our previous study (Robert et al. Science 2018), performed under similar conditions with regards to strains and growth medium, revealed that spontaneous mutations that substantially affect fitness are rare. The purification of rare deleterious mutations by selection in our MA experiments should therefore not significantly affect the estimation of mutation rate. In agreement with this assumption, with our protocol we obtain for a MMR-deficient strain a similar mutation rate as obtained by others with the classical protocol (our results : 0.17 per generation, Lee et al. PNAS 2012 : 0.15 per generation). To further validate our assumption, we performed numerical

simulations of MA experiments in liquid cultures (as in our protocol), where population size varies from 10 to 5×10^9 . Mutations were generated according to a Poisson process, with the rate that we experimentally estimated, and their effects on growth rate followed a Beta distribution compatible with our previous estimation (with 1% of lethal mutations; Robert et al. Science 2018). We found that the bias due to natural selection impacts mutation rate estimation by less than 5%, which is smaller than the precision of our estimation (see Figure 3b). To clarify this point in the manuscript we introduced a dedicated discussion in a section within SI (section 2.2.5).

Reviewer 2

Question 1: It would help to have a cartoon that depicts the logic of the assay the authors are using. If one is away from the field, it takes quite some work through the text to make oneself an image of the elegance of the assay.

Answer 1: We agree. To highlight the rationale of our approach and enhance comprehension we expanded Figure 2D of the original manuscript, separated it from the other panels, and created a new figure, fully dedicated to illustrating the logic of our assay (Figure 2).

Question 2: The authors state that the inter-arrival times of long-lived foci in “mother cells”, which represent unrepaired errors, are not exponentially distributed. This indicates that the error production process is no longer Poissonian, which is compelling in light of the authors’ observations that there is no effect on repair efficiency in aging cells. In figure 5C, three examples of this non-Poissonian behavior are shown. Of these experiments, the green trace (experiment 3) seems to display a strikingly distinct behavior from the other two.

Answer 2: Reviewer 2 is concerned about the differences between the three curves plotted in Figure 5C of the original manuscript, which present data from replicate experiments. To address the reviewer’s concern, we constructed confidence intervals for these curves, using a bootstrap method, as explained in Supplementary text section 2.1.14. In brief, we performed resampling of the data for each dataset, involving sampling with replacement, maintaining the sample size as in the original dataset, and computed the CDF as outlined in Supplementary text section 2.1.13. 500

resampling were performed, leading to 500 CDFs. Figure S12 shows the interval corresponding to 95% of these curves. These confidence intervals confirm that the data is incompatible with the theoretical CDF corresponding to a Poisson process. Moreover, the analysis shows that the differences observed between the three experiments are not statistically significant.

Question 3: In context of the data presented in figure 4C, where the authors show the aging-independence of the repair efficiency, the data in figure 5C seem incomplete. It would help that the authors add two plots to figure 5, showing the absolute time-trace over replicative age (as opposed to the cumulative distribution function) of both the short-lived foci arrivals as well as specifically the long-lived foci arrivals. This would be valuable information for any reader.

Answer 3: We agree with the reviewer that the rate of short-lived foci and of long-lived foci as a function of age is a relevant information. This information was included in the original manuscript in Supplementary Information (Supplementary text 2.1.11 and Figure S11, now Supplementary text 2.1.12 and Figure S10), but was incorrectly cited. We apologize for the confusion caused by this error and we corrected it in the revised manuscript. Note that the data in this figure and Figure 4C of the original manuscript (now Figure 5b) is a “time-trace” (i.e. a variable plotted as a function of time/age). In contrast, Figure 5 (now Figure 6) does not show a time-trace but a probability density (i.e., equivalent to a histogram), presented in a “cumulative” way (i.e., the curve is the integral of the density/histogram). The data in Figure 4 and 5 (now Figures 5 and 6) are therefore of a very different nature.

Question 4: Furthermore, it is not clear from the text whether the non-random mutagenesis is evident in the whole population at different times, or whether this phenotype is only evident in a small subpopulation of the growing cells. While this phenomenon is compelling enough to warrant a dedicated investigation, I recognize that additional experiments in this direction are outside the scope of this work. However, for this work it is critical for the authors to clarify this observation in the text, as it goes directly to their conclusion that the origin of single nucleotide mutations is driven by the rate competition between repair efficiency and methylation. It would be helpful to illustrate this observation with a cartoon.

Answer 4: We are not sure we understand the remark of the reviewer which makes a difference between the “whole population at different times” and “a small population of the growing cells”. We believe that his concern may stem from a lack of details in our explanation of the experimental setup and structure of the data, as pointed by reviewer 1. We added such details in the introduction of the revised manuscript and we also provide a new panel in **Figure 6** (previously Figure 5) showing the arrivals of replication errors in several mother cell lineages (**Figure 6b**). We believe that the new figure clarifies the structure of the population, involving hundreds of independent mother cell lineages followed through 50-100 generations. It also depicts the time-scales of both foci arrivals and cell divisions, which we think should be relevant for addressing the reviewer’s concern. As stated in the main text, the non-poissonian dynamics is visible on **Figure 6d** as an excess of short inter-arrival times (typically $< 1 - 2$ generations). To clarify the implication of this result we added the following sentence at the end of the last paragraph of the corresponding results section: “More precisely, **Fig. 6d** shows that short inter-arrival times between mutations (< 40 minutes) are over-represented, indicating the presence of transient states of decreased repair efficiency and therefore elevated mutation rate, lasting a few generations (typically $< 1 - 2$ generations).” The elevated mutation rate is therefore the property of a single lineage (i.e., single microchannel) in a small window of time. We also followed the suggestion of the reviewer and illustrated the competition between repair and methylation with a cartoon (**Figure 4c**).

Question 5: Regarding the seventh subsection of the Results section (Repair failures are not due to...): the authors state that the filamentous phenotype they observe indicates the induction of some kind of stress response within the cell (e.g. SOS response), but that they do not observe a change in the error repair rate between filamentous and non-filamentous cells. However, a primary result from the study of stress-induced mutagenesis¹ is that two of the primary stress responses of *E. coli*, namely the SOS response and rpoS-mediated stress response, are associated with both the up-regulation of error-prone polymerases, polIV and polV, as well as the down-regulation of MMR machinery. I would expect to see an increase in the number of unrepaired errors, or at the very least in the number of total foci in these cells. I am surprised that the authors observe no difference at all in the filamentous cells. The authors should include the data about total foci observed in the filamentous cells, presented in comparison to those observed in the unstressed cells. Furthermore, it would be quite valuable if the authors were able to further clarify the stress-response phenotype

of the filamentous cells, as it is currently unclear in the text.

The link between stress responses and single base pair mutations represents an interesting research direction. Two possible further experiments would be to make the same observations of short and long-lived foci in a strain of *E. coli* that has an inducible SOS response, and in wild-type cells growing in stressful conditions. However, these would be outside the scope of the paper as it stands.

Answer 5: We agree that the results mentioned by reviewer 2 raise some questions. As reviewer 2 points out, SOS induction through DNA damage (Jaszczur et al., Biochemistry, 2016) or genetic modification of the *recA* gene in *E. coli* results in an increased mutation rate (Niccum, DNA Repair, 2020). However, it's important to note that filamentation does not conclusively indicate SOS induction, as cases of SOS-independent filamentation have been documented (Kawarai, Applied Microbio Tech 2004). Moreover, in SOS-induced cells, the *sulA* gene, responsible for the filamentation phenotype, is expressed prior to the *umuC* and *umuD* genes responsible for almost all the mutagenesis that accompanies the SOS response (Watanabe-Akanuma et al., Mut Research, 1997; Jaszczur et al., Biochemistry, 2016). Consequently, without an examination of the expression profile of SOS genes in filamentous cells, the expectations regarding error and mutation rates in these cells remain uncertain. We fully agree with reviewer 2 that addressing of this issue is beyond the scope of our work, particularly as it is not integral to the main conclusions. Therefore, we have excluded this result from our revised manuscript (Figure 4B in the original manuscript). This decision is in line with the suggestion of reviewer 1, who also expressed concerns about the clarity of this result. The main text and supplementary information have been modified accordingly.

Question 6: Is there a bias in the spectrum of errors that are repaired versus those that are unrepaired? As the authors have determined that error repair is governed by the physico-chemical constraints of the rates of error detection and DNA methylation, understanding whether and how these constraints impose bias on the types of errors that are repaired would be valuable for understanding the repair kinetics. Such analysis could be achieved simply by examining data from the whole genome analyses for both WT and repair-deficient strains without any further experiments.

Answer 6: Our work reveals that: i) many spontaneous mutations stem from replication errors that are detected by MMR but not repaired and ii) many MMR failures are due to DNA methylation of GATC sites occurring before the recruitment of MutH by MutL. Since the methylation status of GATC sites and the availability of MutH do not depend on the nature of the error, the repair failures due to competition between DNA methylation and DNA cleavage are not expected to be error-dependent. Reviewer 2 suggests exploring genome-wide analyses of the mutation spectrum (in WT versus MMR- cells) as done in Lee et al., PNAS (2012) and Niccum et al., Genetics (2018) to further investigate this matter. Note that although we found that many failures occur at the repair step, we cannot be quantitative and estimate the fraction of failures occurring at the detection step. The method suggested would therefore not allow concluding on the error-dependence of repair efficiency because it can not discriminate between repair failures on detected errors and failures occurring at the detection step.

Minor Points

i. At the end of the third paragraph of the Discussion section, the authors indicate that their results suggest an evolutionarily conserved time constraint on the action of mismatch repair systems that may be universally significant. This observation has important consequences for our understanding of the origins of mutations, but is only treated superficially here. It would be a valuable addition to this paper for the authors to support this observation with a longer, literature-based discussion about the evolutionary context of MMR systems and the methylation machinery.

- We agree with this point. Therefore, we have included in our revised manuscript a more detailed discussion regarding the origin of mutations in relation to the evolutionary aspects of MMR systems. (page 9, lines 365-379).

“ Although MMR system is highly conserved across various organisms, the majority of prokaryotic and eukaryotic species do not rely on DNA methylation to guide repair to the newly synthesized DNA strand. In these organisms, the strand discrimination signal has not been conclusively identified but it is commonly assumed to involve replication-associated strand discontinuities and nicks. In *S. cerevisiae*, a model organism for non-methyl-directed MMR, a repair window of approximately 10 to 15 minutes has been reported (Hombauer

2011, Science. This observation suggests a time constraint for error correction, similarly to what is observed for methyl-directed MMR. Studies in *S. cerevisiae* have additionally demonstrated that overexpression of DNA ligase I (Cdc9), causing premature ligation of replication-associated nicks, leads to an elevated mutation rate and the accumulation of fluorescent foci of the MutL homolog, Pms1 (Reyes, Curr Bio, 2021). Conversely, reducing or delaying Cdc9 expression prolongs the temporal window in which loci can undergo MMR (Reyes, Curr Bio, 2021). This indicates that, as reported here for methyl-directed MMR, the effectiveness of non-methyl-directed MMR can be modified by modulating the temporal window available for repair. In the light of our results this suggests that an evolutionarily conserved time constraint on MMR might be setting a limit on its efficiency and explain the origin of an important part of spontaneous mutations universally.”

ii. In figures 2A and 3B, the arrangement of the distributions of foci lifetimes along a diagonal “z”-axis is confusing and detracts from the otherwise compelling point made by these figures. I suggest that these plots be rearranged into a two-dimensional figure so that the “x”-axes of the distributions are aligned with one another.

- We modified the figures according to the reviewer’s suggestion.

iii. In the second paragraph of the third Results subsection, the authors state that the observed average lifetime of the fluorescent foci is forty seconds, which corresponds to the timescale of the MMR repair activity. However, it is not clear from the text if this average time includes the minority of long-lived foci, which correspond to unrepaired errors.

- We agree that it was not clear in the previous manuscript and we thank the reviewer for pointing that out. In order to clarify this point, we i) re-ordered the different paragraphs (the result section dealing with the kinetics of repair is now after all the results that validate the method and describe the short-lived and long-lived foci), and we ii) added a more detailed description of the estimation of repair time in supplementary (Supplementary text section 2.1.8)

In the short-term experiments used to estimate repair kinetics, the lifetimes larger than 2 minutes represent ~2% of the data, and thus one half of them represent unsuccessful repair (long-lived foci). However, they do not significantly bias the estimation of repair time since excluding the lifetimes larger than 2 minutes or including them has a negligible impact of ~3% on the estimated average repair time.

iv. Please check that throughout the text the Fig numbers correspond to the actual figures, for example on page 7 last paragraph Figure S10 should be S11.

- We checked all the manuscript and corrected the mistakes, including the one pointed by reviewer 2.

Reviewer 3

Question 1: The description related to the data and conclusion relate to Fig. 5 is unclear. First, does the repair efficiency change over time in the same cell? If the three experiments are from one cell per experiment, I will agree with the authors' conclusion. If it comes from multiple cells, I would expect to see similar curves as the population average should be similar. please make this clear before drawing the conclusion "the efficiency of error repair is heterogeneous among isogenic cells".

Answer 1: We modified the last section of the results to clarify the logic of the reasoning and analysis. In addition, we added a panel to the original **Figure 5** (now **Figure 6b**), which represents examples of mother cell lineages followed through time, with both short-lived and long-lived foci arrivals, as well as the length of the cells through time which allows visualizing cell divisions. We think this figure should answer the reviewer's concern by highlighting the relative time-scales of cell division and foci arrivals. A single cell typically accumulates only 1 or 2 mutations between birth and division, which is not enough to investigate fluctuations in a single cell cycle, we thus cannot answer to the reviewer's question "does the repair efficiency change over time in the same cell". Each experiment in **Figure 5** (now **Figure 6**) contains the data of thousands of cells (hundreds of independent microchannels containing mother cells imaged for ~30 hours, i.e. ~70 generations). However, long-lived foci are rare events (~0.01 per generation) and the experiments shown in **Figure 6d** typically contain 100-200 long-lived foci. Due to this limited sample size, the CDFs are noisy. In order to assess the reproducibility and to address the concern of reviewer 3 and a similar concern from reviewer 2, we estimated confidence intervals (new **Figure S12**, Supplementary text section 2.1.14), which show that the CDF of the 3 experiments are not significantly different, but significantly different from the CDF corresponding to a Poisson process.

Question 2: Also the initial jump in Fig. 5C is almost similar to the trend observed in the mutH cells in Fig. 5B, seems to me that there are two processes going on here, one is searching process and one is the repairing process. Based on Fig.5B, I will think that the searching process is likely Poissonian. Then the repairing process is likely non-Poissonian as the authors were measuring the time intervals between the two foci, which depends on the arrival time of the second foci as well

as the repairing time in the first foci. In fact, I will not expect it will follow the Poisson process. I would strongly recommend the authors put on a simulation to recapitulate the dynamics in silico. Directly using the non-Poissonian behavior to conclude that “the error repair is heterogeneous” is not convincing

Answer 2: We believe that the reviewer’s concern stems from a misinterpretation of our analysis, probably due to a lack of details in our explanations, so we modified the description of the analysis in the last result section to make it clearer.

As depicted in **Figure 6a**, the inter-arrival time is the time between the first frame where a focus is detected and the first frame where the following one is detected. Therefore, it does not depend on the number of frames where the first (or second) focus is visible, i.e., on the time taken by the repair process. As an example, an inter-arrival time can be 0 if two foci are first detected at the same frame, corresponding to two replication errors produced within a <2min delay. In WT cells, 99% of errors are repaired, meaning that detection by MutS and subsequent repair of a newly created mismatch occurs within the time window of hemimethylation, i.e., ~2 minutes. We found that repair typically takes ~40-50s, thus the “searching time” is typically less than 1-2 minutes. In contrast, the mean inter-arrival time between all fluorescent foci is typically ~20 minutes. Therefore, inter-arrival times in MMR deficient cells, such as analysed in new Fig. 6c, represent mainly the time between the creation of two successive replication errors, the contribution of MutS searching time being negligible. Likewise, for long-lived foci in WT (new Fig.6d), where the inter-arrival times are much longer (~25h), the contribution of MutS searching is also negligible and the inter-arrival time is the time between the arrival of two replication errors on which repair fails and that will lead to 2 successive mutations. It cannot be that “there are two processes going on here, one is searching process and one is the repairing process”. Note also that these foci are not repaired (long-lived foci mark unsuccessful repair).

As now explained in the main text of the revised manuscript, the mathematical process describing mutation occurrence in a WT context if repair efficiency is constant is a compound poisson process (poissonian dynamics of replication errors) with Bernoulli increments (Bernoulli trial for the success or failure of the repair process), which is equal to a Poisson process with a rate which is the product of the replication error rate by the probability of successful repair. Our finding that mutations have a non-poissonian dynamics therefore demonstrate that repair efficiency is not

constant. This explanation is now clearly stated in the revised manuscript.

Minor Points

i. In Fig. S2A, the authors presented a control experiment to rule out the potential effects caused by photobleaching. The photobleaching of three conditions (7.5s, 15s, 30s) were shown in this plot. It would be better if the authors could include the longer-time conditions (e.g., 2min) used in this manuscript.

- We agree and thank the reviewer for this suggestion. We have included the plot corresponding to the long-term experiments in **Figure S2b**.

ii. Please provide a summary of some basic metrics regarding the foci detection. For example, how many foci can be detected in each cell on average? How many short-lived foci and long-lived foci were detected in each single cell?

- We agree with the reviewer that such summary is relevant. It was already in the supplementary text of the previous manuscript but hard to find since it was not cited in the text. We now cite this table clearly in the last paragraph of the introduction (line 90).

iii. Regarding the long-term monitoring of MutL foci experiment (Fig. 5), it will be helpful to include a more detailed experimental scheme, like Fig. S12A, in the main figures.

- We've added two new panels to Figure 5 (now **Figure 6**) to clarify the experimental scheme and analysis.

Reviewer #1 (Remarks to the Author):

Bene et al. use microfluidic fluorescence in a mother machine to determine the nature of MMR failures through YFP-MutL fluorescent imaging tracking. The work is novel and interesting, it allows for detection of enzyme kinetics of repair in *E. coli* and uses advanced microfluidic technology to do so, expanding upon their previous work.

With regards to the responses from the original review:

- 1) The authors expand upon the use of the mother machine describing the process for readers. The added paragraph in the first section is sufficient for description and necessary to depict the logic.
- 2) The authors remove analysis of YFP-mutL over time when describing cells with different segmentation times which did not seem appropriate given the data that they had. Figure 4 and 5 have been updated to present relevant data.
- 3) The authors describe the differences between MA experiments and their methodology and provide a supplemental section describing the differences and the possible consequences.

The reviewers have done a sufficient job revising their manuscript.

Reviewer #2 (Remarks to the Author):

To the editor and manuscript authors,

I have read the authors' response to the comments from myself and reviewers 1 and 3. The major revisions include the removal of the results and discussion concerning mutation-indicating fluorescent foci appearing in filamentous cells. In the original manuscript, these results were not treated with sufficient depth and indeed fell outside the scope of the rest of the paper (a sentiment that was seemingly shared by reviewer 1). In my view, this exclusion makes the resulting manuscript clearer and more focused.

The authors have sufficiently clarified my confusion regarding the origin of elevated mutation rates in the population (Reviewer 2, answer 4), through expansion of the text and the inclusion of a new figure panel in figure 6. For maximum clarity in the text, I suggest that the following line from this section of the authors' reply be added directly to the text in the last paragraph of the results section (lines ~300 - 313): "The elevated mutation rate is therefore the property of a single lineage [...]".

In addition, the authors added a helpful explanation of the consequence of observing non-Poissonian mutation dynamics, i.e. that the emergence of mutations is in fact a compound Poisson process with Bernoulli increments representing the repair process and that non-Poisson dynamics indicate variability of repair efficacy. This addition comprehensively addresses one of the major questions I had posed in my comments.

The authors also added an expanded discussion on the possibility of evolutionarily conserved time constraint that affects the behavior of mismatch repair (MMR) systems and supporting their conclusions with the example of a non-methyl-directed MMR system from *S. cerevisiae*.

I am satisfied that the authors have sufficiently addressed the concerns that I brought up about their manuscript and I recommend that this version be accepted for publication without further revisions.

Reviewer #3 (Remarks to the Author):

The clarity of the manuscript has been significantly improved. I will recommend the acceptance of the revised manuscript.